# A Rare Triploid Involving the Coexistence of Glioblastoma Multiforme, Arteriovenous Malformation and Intracranial Aneurysm: Illustrative Case and Literature Review

**DOI:** 10.3390/medicina59020331

**Published:** 2023-02-10

**Authors:** Chi-Ruei Li, Se-yi Chen, Meng-Yin Yang, Wen-Yu Cheng, Chiung-Chyi Shen

**Affiliations:** 1Department of Neurosurgery, Neurological Institute, Taichung Veterans General Hospital, Taichung 40705, Taiwan; 2Department of Neurosurgery, Chung-Shan Medical University Hospital, Taichung 40201, Taiwan

**Keywords:** glioblastoma multiforme (GBM), arteriovenous malformation (AVM), internal carotid artery (ICA) aneurysm, digital subtraction angiography (DSA)

## Abstract

The coexistence of glioblastoma multiforme (GBM) and arteriovenous malformation (AVM) is rarely reported in the literature. According to the present literature, these GBM or glioma-related vascular malformations may present simultaneously in distinct regions of the brain or occur in the same area but at different times. So far, these distinct hypervascular glioblastomas have been described but are not classified as a separate pathological entities. Considering their heterogeneity and complexity, all the above mentioned cases remain challenging in diagnosis and therapeutic modality. Likewise, there is a paucity of data surrounding the simultaneous presentation of GBM with intracranial aneurysms. In the literature, the independent concurrence of these three intracranial lesions has never been reported. In this article, we present a case who suffered from intermittent headaches and dizziness initially and further radiographic examination revealed an internal carotid artery (ICA) aneurysm that occurred in the patient with coexisting GBM and AVM. Surgical intervention for tumor and AVM removal was performed smoothly. This patient underwent endovascular coiling for the ICA aneurysm 4 months postoperatively. In addition, we also review the current literature relating to this rare combination of medical conditions.

## 1. Introduction

The coexistence of glioblastoma multiforme (GBM) and arteriovenous malformation (AVM) is rarely reported in the literature, and at times the term “angioglioma” is used to describe hypervascular glial cell tumors. Tunthanathip et al. [1] made a review containing 67 cases since 1969; however, most of them were glial cell tumors surrounded by vascular lesions which were different from the present case. Additionally, there is a paucity of data surrounding the simultaneous presentation of glioblastoma multiforme (GBM) with intracranial aneurysms. In the English-language literature, the concurrence of these three intracranial lesions has never been reported.

Regarding the present case, the pathological entity may resemble the abovementioned “angiogliomas” to some degree although they are usually hypervascular low-grade lesions with a favorable prognosis. According to the present literature, these GBM or glioma-related vascular malformations may present simultaneously in distinct regions of the brain or occur in the same area but at different times. So far, these distinct hypervascular glioblastomas have been described but have not been classified as a separate pathological entities. Considering their heterogeneity and complexity, all the above mentioned cases remain challenging in diagnosis and therapeutic modality.

In addition to the clinical illustration of the rare combination of GBM, AVM and ICA aneurysm, we also searched the literature focusing on pathogenesis, especially in molecular investigation, and discuss whether there is a correlation between these three distinct but simultaneously existing lesions.

## 2. Case Presentation

A 44-year-old female was presented to our hospital experiencing intermittent headaches and dizziness. Furthermore, she had experienced a change in consciousness three times in the previous month. Her neurological examination was normal except for slight paresthesia in her right hand. Computed tomography demonstrated enhancement with a peri-focal edema lesion over the right temporo-parietal area Figure 1A. Additionally, another lesion with focal heterogeneous enhancement was noted over the right parieto-occipital region Figure 1B.

Magnetic resonance imaging (MRI) demonstrated a well enhanced mass, 1.1 cm × 2.3 cm in size, with a peri-focal edema in the right temporal lobe Figure 2A. In addition, surrounding and tangled dilated vessels in the right temporo-occipital area were noted on T1-weighted with gadolinium Figure 2B. Computed tomographic angiography (CTA) also confirmed these lesions Figure 3. Upon additional CTA study, the presence of arteriovenous malformation (AVM) nidus at the right temporal-parietal occipital region with arterial feeding from the right middle cerebral artery (MCA) and posterior cerebral artery (PCA), which then drained to the right transverse-sigmoid sinus via an engorged drainage vein, was noted.

We arranged a navigation-guided right temporo-parieto-occipital craniotomy under general anesthesia. We planned to remove the AVM first and then remove the temporal lesion. We left the observed unruptured aneurysm since it was difficult to clip with the same approach. During the operation, the resection of the temporo-occipital AVM was performed first. Then, we aimed to achieve the gross resection of the temporal lesion. During the tumor removal procedure, the lesion was noted at the right temporal area with dural adhesion, while a cystic part with straw-colored contents was also noted. The patient recovered well and was discharged on the postoperative day 11.

Histopathological analysis of the tumor revealed glioblastoma (WHO grade IV), with increased cellularity, cellular atypia, and frequent mitotic figures. Glomeruloid microvascular proliferation and tumor necrosis were also found Figure 4 and Figure 5. Immunohistochemical (IHC) staining revealed a positive stain of glial fibrillary acidic protein (GFAP), isocitrate dehydrogenase-1 (IDH-1) and alpha thalassemia/intellectual disability X-linked gene (ATRX) Figure 6A–C. A wild-type pattern presentation of p53 protein was also noted Figure 6D. In addition, the other specimen revealed a mixture of abnormally muscularized arteries without continuity and a complete three-layer structure under Masson’s trichrome stain Figure 7. Furthermore, ambiguous vessels exhibiting both artery and vein characteristics that were compatible with AVM under Verhoeff–Van Gieson stain Figure 8.

Following discharge, she underwent concurrent chemoradiotherapy with a total dose of 6900 cGy given in 30 fractions in conjunction with temozolomide (TMZ) at 75 mg/m^2^ daily. Upon a follow-up visitation 6 months post-operation, apart from previous neurologic deficits, she remained in good condition. Surveillance imaging showed no disease progression in the past 18 months after the surgery. We performed digital subtraction angiography (DSA) 4 months later and found no evidence of AVM, but the ICA aneurysm revealed only a sparse change in size without a sign of rupture Figure 9. After a detailed discussion with the patient and her family members, they decided to treat the aneurysm with endovascular intervention.

## 3. Discussion

Simultaneous presentation of vascular lesions and glioblastoma are rarely encountered, while the coexistence of AVM, glioblastoma and an intracranial aneurysm is never reported. Tunthanathip et al. [1] made a review containing 67 cases since 1969 however most of them were glial cell tumors surrounded by vascular lesions which were different from the present case. These coexistences have been found in both low-grade and high-grade gliomas. The common glial cell tumors are GBM, pilocytic astrocytoma, and oligodendroglioma. Regarding their histological features, GBMs are usually found with endothelial cell proliferation and a glomeruloid appearance, while pilocytic astrocytoma and oligodendroglioma have been found with other characteristics of abnormal vessel pattern.

The term “angioglioma” was originally used by Russell and Rubinstein to describe a tumor possessing characteristics of angiogenic and glial origin, including gliomas in combination with either venous malformations, cavernous angioma or arteriovenous malformations [2,3]. These angioglioma usually appeared in close vicinity to the abnormal vascularity and glioma. The pathogenesis of these angioglioma was uncertain. McKinney et al. [3] provided evidence regarding the de novo formation of an AVM within an anaplastic oligodendroglioma in a patient and proposed the hypothesis that a hyperangiogenic stimulus provided by the tumor may have initiated the development of the AVM. A similar hypothesis was provided in previously proposed studies that claimed an increase in blood flow due to a higher blood supply to a malignant glioma may induce secondary changes in the arterial wall, thus stimulating the formation of aneurysms [4,5]. According to studies performed regarding C-X-C chemokine Receptor type 4 (CXCR4), Huang et al. [6] demonstrated the association between CXCR4 with angiogenesis and tumor progression. These receptors are upregulated by Hypoxia-inducible Factor-1 (HIF-1). In addition to HIF-1, vascular endothelial growth factors (VEGFs) also had a prominent influence in the upregulation of CXCR4 receptors in those GBM cases [7].

In vascular malformation, it has been shown that VEGF expression plays a fundamental role in mediating angiogenesis and may also play an important role in both AVM development and progression. In the presence of VEGF, Angiopoietin-2 induces pericytes to dissociate themselves from existing vessels and facilitates the extravasation of other pro-angiogenic factors. These events finally result in the formation of new blood vessels [8]. On the other hand, another hypothesis has implied that the neoplastic changes in perivascular glial tissue was secondary to vascular malformation. This hypothesis that intracranial vascular malformation may play a role in tumorigenesis has also been supported by earlier studies [9,10,11].

For the coexisting intracranial aneurysm, we have reviewed the current literature about the coexistence of aneurysms and glioma. Ali et al. [12] showed 19 case reports of a combination of aneurysms and GBMs in a 2015 study. For these patients, 43% of aneurysms were unrelated to the tumor, 16% of the aneurysms were pseudoaneurysms or dissecting aneurysms, and 8% were flow-related aneurysms. The study further developed strategies for these coexisting lesions. For the aneurysms which are located remotely from the tumor or located on a vessel that might be supplying the tumor, tumor resection or initial adjuvant therapy should be performed with a priority to eliminate mass effect and achieve cytoreduction. If the aneurysm is located on the ipsilateral side to the tumor, it may be clipped in the same surgery or could be clipped or coiled later. Nowadays, treatment for patients with coincident malignant glioma and intracranial aneurysm remains controversial and requires special consideration. The treatment strategy requires the spatial relationship between the aneurysm and malignant glioma and overall prognosis of the patient. Most of all, the more symptomatic lesion should be treated first.

Regarding the present case, we expected that the supra-clinoid aneurysm would reduce in size after removal of the ipsilateral AVM and glioblastoma due to intracranial hemodynamic changes and fewer angiogenesis-stimulation factors such as VEGF from the tumor. A similar hypothesis was addressed in the current literature and demonstrated good outcomes in cases where intracranial aneurysms and brain tumors simultaneously coexisted [13,14], but unfortunately the postoperative DSA revealed only sparse change in the right ICA aneurysm in the present case. Therefore, the patient still needed to receive intervention involving flow diverter stent placement.

In hindsight, a pre-operative DSA or utilization of a hybrid operating room with intra-operative DSA might provide a safer strategy facing lesions with complex angioarchitecture. Secondly, more extensive genetic/biomolecular investigation should be performed. For example, the IHC stain could include HIF-1 and VEGF concerning the overexpression status. For the lesions with hypervascularity, these factors play important roles in angiogenesis mediation and the IHC stain data could provide us with a more apparent correlation between these coexisting lesions. Furthermore, an analysis of the methylation status of the O6-methylguanine-DNA methyl-transferase (MGMT) gene promoter might provide us with significant information in predicting patient survival and response to chemoradiotherapy with the alkylating agent [15,16,17].

## 4. Conclusions

The coexistence of vascular malformation, aneurysm, and malignant glioma has been rarely reported. The treatment plan is usually challenging and requires multiple modalities for the complexity of each lesion. The correlation between these lesions remains uncertain and needs more genetic and molecular studies. With regard to our case, the pre-surgical image survey provided us with much support towards making critical decisions. Surgery surrounding the removal of simultaneously coexisting AVM and GBM is always a challenging task due to the potential risk of massive intraoperative bleeding.

## Figures and Tables

**Figure 1 medicina-59-00331-f001:**
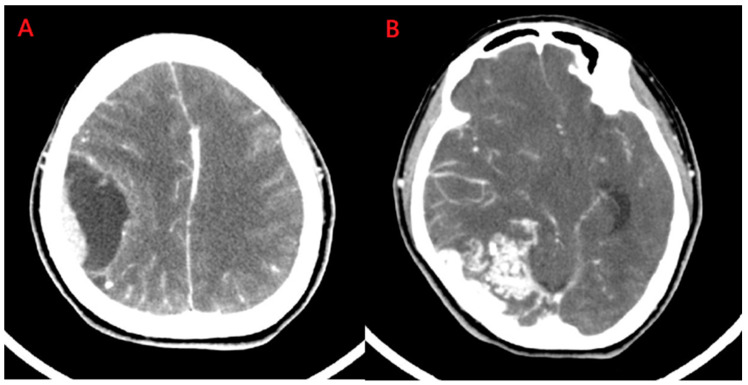
Computed tomography of brain. (**A**) The lesion with perifocal edema was seen in the right temporo-parietal area; (**B**) Another lesion with focal heterogeneous enhancement and a surrounding multilinear contrast structure with a lower density and calcified spots was noted in the right parieto-occipital region.

**Figure 2 medicina-59-00331-f002:**
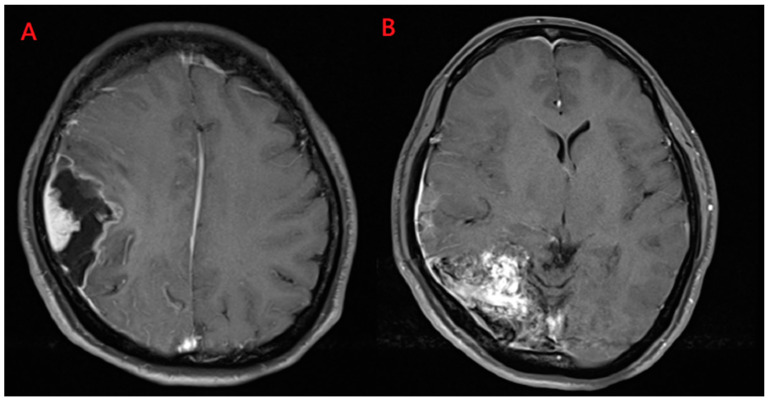
Magnetic resonance imaging scans. (**A**) Heterogeneously enhanced mass, 1.1 cm × 2.3 cm in size, with perifocal edema in the right temporal lobe was noted; (**B**) Surrounding and tangled dilated vessels in the right temporo-occipital area were observed on T1-weighted images with gadolinium.

**Figure 3 medicina-59-00331-f003:**
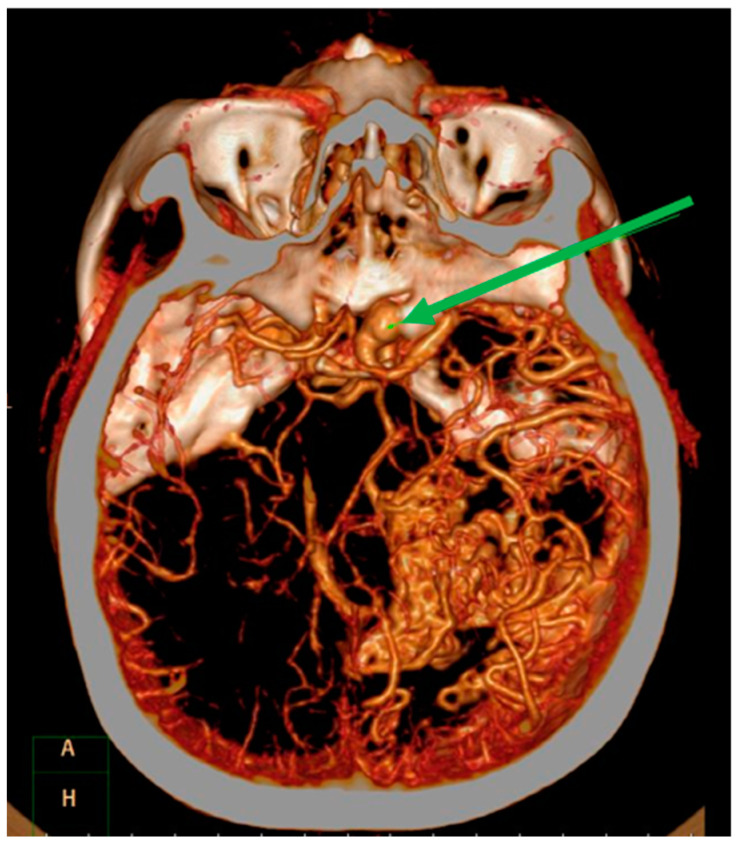
Computed tomography angiography. Imaging revealed focal heterogeneous enhancement with a surrounding multilinear contrast structure, and a lower density and calcified spots in the right temporal-parietal occipital lobe. The green arrow shows an ipsilateral supraclinoid internal carotid artery aneurysm.

**Figure 4 medicina-59-00331-f004:**
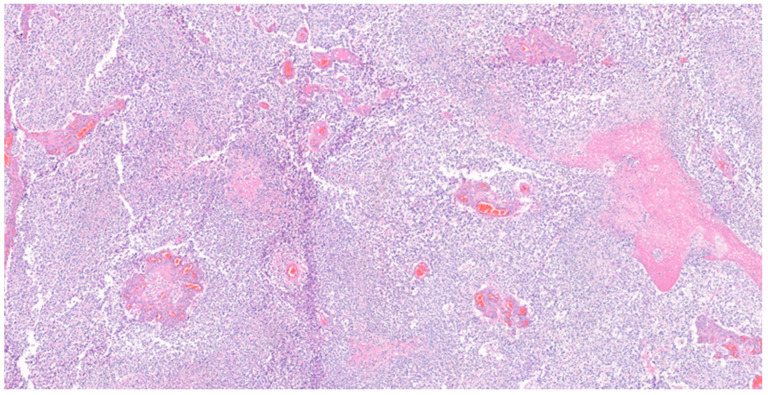
Histological examination of the tumor. Necrosis and high cellularity were noted. Numerous blood vessels of varying sizes were also noted within the tumor. Hematoxylin and eosin stain, magnification 50×.

**Figure 5 medicina-59-00331-f005:**
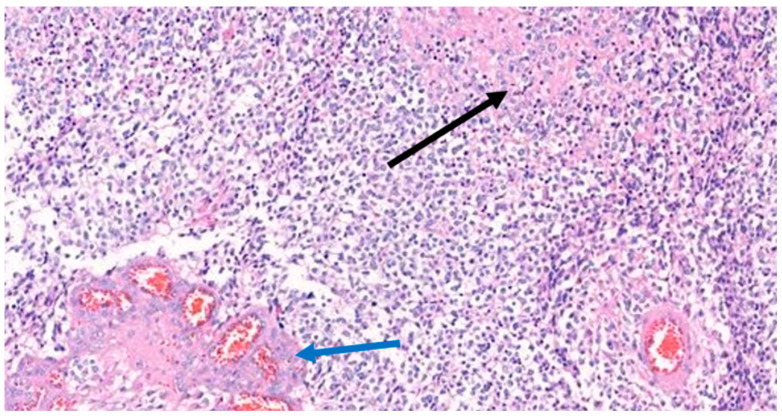
Histological examination of the tumor. The neoplastic cells had nuclear enlargement where hyperchromatic nuclei could be observed. The black arrow shows geographic necrosis, and the blue arrow shows microvascular proliferation. Hematoxylin and eosin stain, magnification 200×.

**Figure 6 medicina-59-00331-f006:**
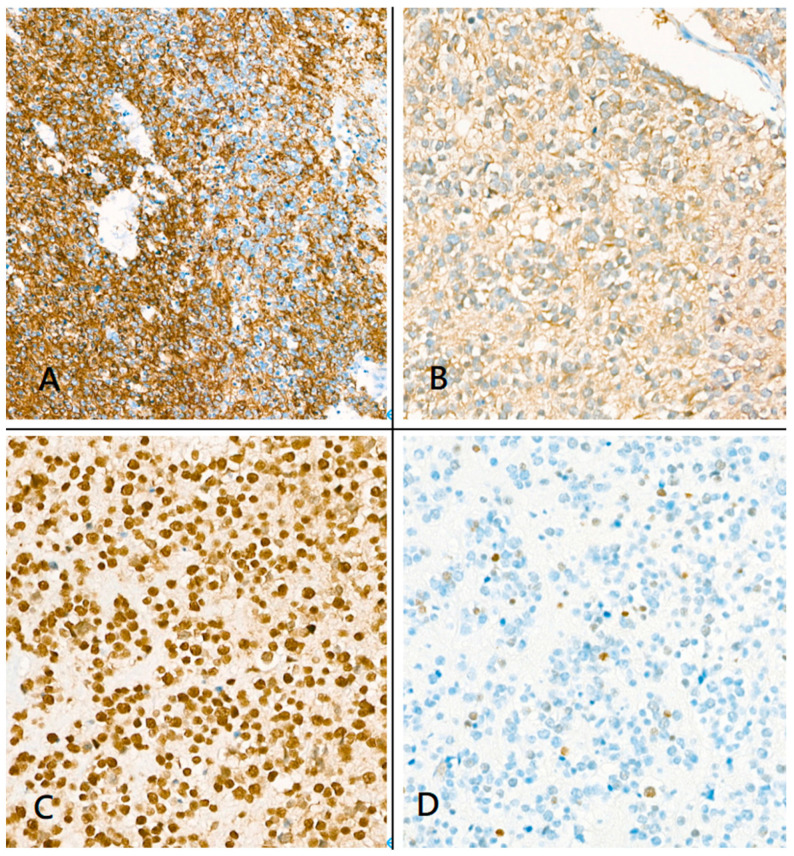
Immunohistochemical staining. (**A**) Immunohistochemistry showed that the tumor was positive for glial fibrillary acidic protein; (**B**) Isocitrate dehydrogenase-1-positive staining; (**C**) Alpha-thalassemia/intellectual disability X-linked-positive staining; (**D**) p53 staining showed a wild-type pattern.

**Figure 7 medicina-59-00331-f007:**
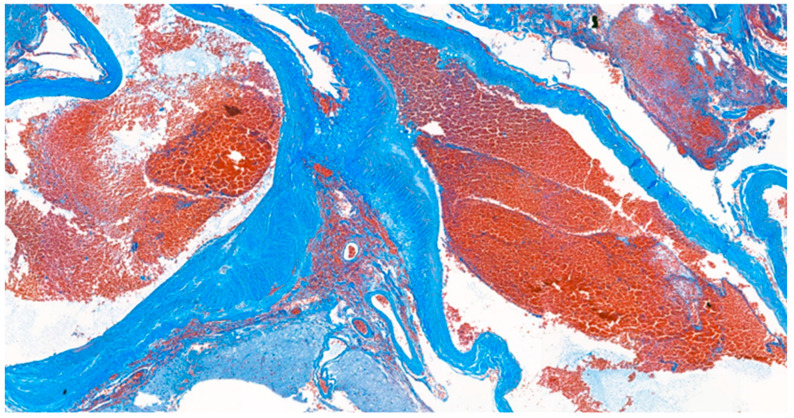
Masson’s trichrome stain. The histological stain revealed abnormally muscularized arteries.

**Figure 8 medicina-59-00331-f008:**
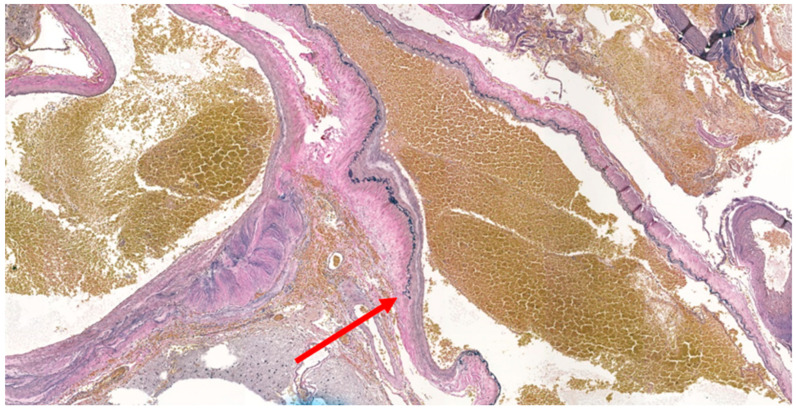
Verhoeff–Van Gieson stain for arteriovenous malformation. Staining showed discontinuity of the elastic fibers, which revealed ambiguous vessels exhibiting characteristics of both arteries and veins, consistent with an arteriovenous malformation (red arrow).

**Figure 9 medicina-59-00331-f009:**
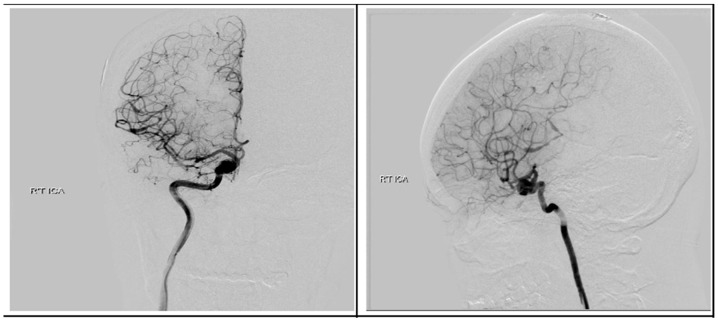
Postoperative digital subtraction angiography. Angiography revealed a right supraclinoid internal carotid artery aneurysm.

## Data Availability

No new data were created or analyzed in this study. Data sharing is not applicable to this article.

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
