# Peer review of "A Rare Triploid Involving the Coexistence of Glioblastoma Multiforme, Arteriovenous Malformation and Intracranial Aneurysm: Illustrative Case and Literature Review"

_medicina, 2023, doi:10.3390/medicina59020331_

Round 1

Reviewer 1 Report

Indeed, the manuscript shows the rare cases with the coexistence of glioblastoma multiforme (GBM) and arteriovenous malformation (AVM) and intracranial aneurysms.

However, I think it lacks important messages with some scientific evidence for the readers.

There should have been any genetic analysis or biochemical examinations required.

 Also, the discussion seems to be redundant and irrelevant. 

Author Response

For this special combination of GBM, AVM and aneurysm, more detailed genetic analysis or biochemical examinations should be performed indeed. To make more precise therapeutic modality, more detailed biochemical analysis such as predictive factor MGMT gene promoter methylation status for the chemotherapy effect will be performed.

Reviewer 2 Report

This manuscript presented the rare concurrent coexistence of vascular malformation, aneurism, and glioblastoma. It provided detailed information regarding the clinical course of the disease supported by the descriptive figures of imaging and IHC analyses. In conclusion, the authors emphasized the importance of the pre-surgical image survey in making clinical decisions and the plan of treatment. Thus, this manuscript, in its current form, presents valuable information worth publishing. 

The nature of the correlation between these lesions needs extensive genetic/molecular investigation, as the authors admitted. I am aware that such an investigation exceeds the limits of presenting case study. However, such a correlation could be more apparent if the authors provided some additional information. They mentioned the role of HIF-1 and (especially) VEGF in mediating angiogenesis. Given that, I would ask them a few questions:

  1. Is there a possibility of providing the IHC data concerning their overexpression?  
  2. Did you analyze the MGMT promoter methylation status of the patient?

Minor typographical/linguistical corrections should be performed:

Line 107 – remove this line

Line 108 – remove this line

Line 141 – Change “decision making” to “decision-making”

Line 151 – Change “required” to “requires”

Author Response

  1. For the IHC stain, we did not preformed the analysis for HIF-1 and VEGF.  Indeed, the analysis of  HIF-1 and VEGF could provide us more information in these hypervascular lesions.
  2. For this patient, we did not perform the analysis for the predictive factor of favorable survival in glioblastoma patients undergoing chemotherapy with alkylating agents.